# A Novel Role for the Tumor Suppressor Gene *ITF2* in Tumorigenesis and Chemotherapy Response

**DOI:** 10.3390/cancers12040786

**Published:** 2020-03-26

**Authors:** Olga Pernía, Ana Sastre-Perona, Carlos Rodriguez-Antolín, Alvaro García-Guede, María Palomares-Bralo, Rocío Rosas, Darío Sanchez-Cabrero, Patricia Cruz, Carmen Rodriguez, MDolores Diestro, Rubén Martín-Arenas, Verónica Pulido, Pilar Santisteban, Javier de Castro, Olga Vera, Inmaculada Ibáñez de Cáceres

**Affiliations:** 1Epigenetics Laboratory, INGEMM, Hospital La PAZ. 28046 Madrid, Spain; olgacarpe@gmail.com (O.P.); amsp84.as@gmail.com (A.S.-P.); rodriguez.antolin.c@gmail.com (C.R.-A.); alvarogr@ucm.es (A.G.-G.); rorosas89@gmail.com (R.R.); carmenrjnett@hotmail.com (C.R.); vpulido@ucm.es (V.P.); 2Experimental Therapies and novel biomarkers in cancer IdiPAZ. 28046 Madrid, Spain; dario.sc88@gmail.com (D.S.-C.); patriciacruzcastellanos@gmail.com (P.C.); javier.decastro@salud.madrid.org (J.d.C.); 3Instituto de Investigaciones Biomedicas CSIC/UAM. 28029 Madrid, Spain; pilar.santisteban@iib.uam.es; 4Ciber de Enfermedades Raras (CIBERER). Instituto de Salud Carlos III (ISCIII), 28029 Madrid, Spain; maria.palomares@salud.madrid.org (M.P.-B.); rmarenas@salud.madrid.org (R.M.-A.); 5Laboratorio de Genómica Estructural y Funcional, INGEMM. IdiPAZ. 28046 Madrid, Spain; 6Gynaecologic Oncology Unit, La Paz University Hospital, 28046 Madrid, Spain; mdtejeda@gmail.com; 7Ciber de Cáncer (CIBERONC) Instituto de Salud Carlos III (ISCIII), 28029 Madrid, Spain; 8Department of Molecular Oncology, H. Lee Moffitt Cancer Center and Research Institute, Tampa, FL 33612, USA

**Keywords:** ITF2-TCF4, chemotherapy resistance, NSCLC, HOXD9, Wnt pathway

## Abstract

Despite often leading to platinum resistance, platinum-based chemotherapy continues to be the standard treatment for many epithelial tumors. In this study we analyzed and validated the cytogenetic alterations that arise after treatment in four lung and ovarian paired cisplatin-sensitive/resistant cell lines by 1-million microarray-based comparative genomic hybridization (array-CGH) and qRT-PCR methodologies. RNA-sequencing, functional transfection assays, and gene-pathway activity analysis were used to identify genes with a potential role in the development of this malignancy. The results were further explored in 55 lung and ovarian primary tumors and control samples, and in two extensive in silico databases. Long-term cell exposure to platinum induces the frequent deletion of *ITF2* gene. Its expression re-sensitized tumor cells to platinum and recovered the levels of Wnt/β-catenin transcriptional activity. *ITF2* expression was also frequently downregulated in epithelial tumors, predicting a worse overall survival. We also identified an inverse correlation between *ITF2* and *HOXD9* expression, revealing that Non-small cell lung cancer (NSCLC) patients with lower expression of *HOXD9* had a better overall survival rate. We defined the implication of *ITF2* as a molecular mechanism behind the development of cisplatin resistance probably through the activation of the Wnt-signaling pathway. This data highlights the possible role of *ITF2* and *HOXD9* as novel therapeutic targets for platinum resistant tumors.

## 1. Introduction

Although platinum-based chemotherapy still plays an important role in the treatment of many solid tumors, the disease progresses to a platinum-resistant state in a high percentage of the diagnosed cases of non-small cell lung cancer (NSCLC) and ovarian cancer [1,2] which are two of the most deadly cancers plaguing our society, the former accounting for more than 80% of primary lung cancer cases and the latter boasting the highest mortality of the gynecological malignancies worldwide [3]. Cisplatin (CDDP) is a platinum compound widely used in the treatment of solid tumors. It induces apoptosis in cancer cells by binding to the **guanine** base of DNA, at the number **7 nitrogen** (N7) atom of the purine ring and crosslinking DNA [4,5]. However, CDDP also leads to cytogenetic alterations, such as deletions or amplifications of genes involved in tumor progression, metastasis, and drug response [6], which contribute to the development of CDDP-resistance [7,8,9].

In this study, we performed a high-resolution million feature array-comparative genomic hybridization (aCGH) with four NSCLC and ovarian cancer sensitive/resistant paired cell lines, previously reported by our group [10], to explore the chromosomic deletions that differ in the resistant subtypes. We found a common deletion that includes transcription factor 4, *TCF4* (hereafter called *ITF2*). *ITF2* is a downstream target gene of the Wnt/β-catenin pathway that negatively regulates its activity [11,12]. Wnt signaling has been identified as one of the key signaling pathways in cancer, and more recently, also to be involved in drug resistance of primary tumors such as colon or ovarian cancer [13,14]. However, its role driving platinum resistance in NSCLC has not been defined yet, and very little is known about how ITF2 is involved in tumorigenesis. Therefore, further studying the role of *ITF2* in tumor response to chemotherapy may provide new ways to fight the resistance to this popular treatment. 

Here we report the frequent downregulation of *ITF2* in NSCLC patients and cisplatin-resistant cancer cells. Furthermore, we present potential molecular mechanisms, including the Wnt-signaling pathway behind the development of resistance through the action of *ITF2* and affecting the expression of specific genes that might also be used as potential therapeutic targets. 

## 2. Results

### 2.1. ITF2 is Frequently Downregulated by Chromosomal Deletion after CDDP Cell Exposure

We used CDDP sensitive and resistant clones from NSCLC (H23 and H460) and ovarian cancer (A2780 and OVCAR3) cell lines, previously generated by our group, to perform CGH arrays and uncover deletions or amplifications that could explain the Cisplatin-resistant phenotype. Our cytogenetic study showed different genomic alterations in the CDDP resistant subtypes, using the sensitive parental cell lines as a reference genome. We found two deleted regions shared by both tumor types in three out of the four cell lines H23R, A2780R, and OVCAR3R, but not in H460 cells, located on 18q21.2–18q21.31 and 18q21.32, affecting the genes *RAB27B*, *CCDC68*, *TCF4*, *TXNL1*, *WDR7*, and *BOD1P*; and genes *ZNF532*, *SEC11C*, *GRP*, respectively (Figure 1 and Appendix A). We also observed a common deleted region on 2q22.1 that included the gene *LRP1B* in the NSCLC cell lines and an additional common region only shared by the ovarian cancer cell lines on 9q22.33 that included part of the gene *TMOD* (Appendix A).

We selected *LRP1B* and *TCF4 (ITF2)* genes that were completely deleted in the same tumor type or at least three of the four cell lines, respectively (Figure 1A and Appendix A). The deletion of *ITF2* in H23R and A2780R cell lines resulted in a significant loss of *ITF2* expression compared to the sensitive subtypes, validating the results obtained in the arrays CGH (Figure 1B). No changes were observed in OVCAR3 cells for *ITF2* expression or in H23 and H460 cells for *LRPB1* (Figure 1B and Appendix A). 

### 2.2. Restoration of ITF2 Increases the Sensitivity to CDDP and Decreases β-catenin/TCF Transcriptional Activity

Due to the association of *ITF2* with the Wnt/ß-catenin/TCF pathway, we studied the transcriptional activity of the Wnt pathway in H23S/R and A2780S/R cells, in which the cisplatin-resistant phenotypes harbor the *ITF2* deletion. We transfected cells with the Wnt reporters (Super8xTop-Fop vectors) and induced the ß-catenin/TCF4 activity either by LiCl treatment, which inhibits GSK3-ß or by cotransfection with the constitutively stable ß-catenin-S33Y mutant. We observed higher luciferase activity in A2780R cells compared with the parental sensitive ones, indicating increased transcriptional activity of ß-catenin/TCF in response to both pharmacological (Figure 2A) and functional activation of the pathway (Figure 2B). We also analyzed the RNA-seq normalized FPKM values in paired cell lines, confirming the significant decrease of *ITF2* expression in the resistant phenotype and observing an increase in the expression levels of several downstream effectors genes of this pathway such as *DKK1* (*p* = 0.041), *TCF7L1* (*p* = 0.027), and *TCF7L2* (ns) (Figure 2C), being *DKK1* the most evident in terms of FKPM values. These effects were not observed in H23S/R paired cell lines (Figure 2D).

In order to test the role of *ITF2* in cisplatin resistance through the Wnt pathway, we transiently overexpressed *ITF2* cDNA in A2780 and H23 cells. Cell sensitivity to cisplatin was unmodified after ITF2 overexpression in H23 cells as expected, probably due to the constitutive activity of the Wnt pathway already described in this cell line (Appendix A). However, our previous results indicated that A2780 cells seemed to be a reliable cellular model to evaluate changes in the transcriptional activity of the Wnt pathway. In fact, we observed that the overexpression of *ITF2* in A2780R resulted in a significant increase in sensitivity to cisplatin from the dose of 0.5 ug/mL (*p* < 0.01), showing an intermediate phenotype between the resistant and sensitive subtypes (Figure 3A). In addition, *ITF2* restoration induced a dramatic decrease in cell viability (*p* < 0.05) 24 h after transfection compared with the parental resistant cells transfected with the empty vector (Figure 3B). *ITF2* overexpression at 24 and 72 h after transfection was confirmed by qRT-PCR (Figure 3C). Moreover, *ITF2* overexpression recovered the levels of ß-catenin/TCF transcriptional activity observed in sensitive cells (Figure 3D). We also determined *DKK1* expression levels after *ITF2* overexpression as it was the downstream effector gene showing the highest increased expression in the resistant A2780R cells. In fact, its expression was restored in part at 24 h (*p* < 0.05) after *ITF2* overexpression (R-ITF2). Decreased levels of DKK1 were also observed at 72 h in both, semi-quantitative and quantitative assays, without associated significance, probably due to the transient transfection (Figure 3E,F).

### 2.3. The Expression of ITF2 is Frequently Downregulated in NSCLC, Ovarian and Other Epithelial Tumors

To validate our in vitro results, we determined the clinical implication of *ITF2* and *DKK1* expression in NSCLC and ovarian cancer patients. The relative expression of both genes was measured in two cohorts of fresh frozen tumor samples (T), adjacent tissue (ATT) from NSCLC (Table 1), and ovarian cancer patients (Appendix A).

We observed that *ITF2* expression was frequently downregulated in NSCLC and ovarian tumor samples (Figure 4), validating our in vitro data. Fifteen out of 25 tumor samples of NSCLC patients had lower expression of *ITF2* compared to the normal lungs mean (NLM) (Figure 4A). Furthermore, as reported in our experimental data, we observed the opposite expression profile between *ITF2* and *DKK1* in 60% of NSCLC samples. However, this situation was found only in approximately 10% (1 out of 9) of the ovarian cancer samples (Figure 4B). We did not observe differences between ATT and normal lung samples (LC) for *ITF2* (*p* = 0.177) and *DKK1* (*p* = 0.693) in the NSCLC cohort (Figure 4A).

The Kaplan–Meier curves, analyzing the overall survival (OS) according to the median of *ITF2* and *DKK1* expressions, showed that only *ITF2* high expression levels had a trend towards better overall survival in NSCLC patients with high *ITF2* expression levels after 250 days of following up (*p* = 0.1) (Figure 4C and Appendix A). However, the statistical significance of both genes in predicting overall survival was confirmed by analyzing an extended cohort of 1926 lung cancer patients using the Kaplan–Meier plotter online tool, revealing that those patients with high expression of *ITF2* (Figure 4D) and low expression of *DKK1* (Figure 4E) had a significantly better overall survival rate (*p* = 0.016 and *p* < 0.001, respectively). This was validated using the TCGA data set of 487 lung carcinoma (Appendix A). In addition, *ITF2* was found deleted or downregulated in other epithelial tumors, such as esophageal adenocarcinoma (186 patients) or head and neck **Squamous cell carcinoma** SCC (522 patients), where *ITF2* loss also predicted a worse overall survival (Appendix A). 

### 2.4. Identification of Candidate Genes Involved in the Wnt Signaling Pathway through the Analysis of RNA-seq in NSCLC Patients

To further explore the role of the Wnt-signaling pathway in lung cancer tumorigenesis, we performed a whole transcriptome analysis performing RNA-seq on 14 samples, including nine NSCLC samples, six of them with an inverse expression profile between *ITF2* and *DKK1*, Patient 3 (Pat3), Patient 6 (Pat6), Patient 10 (Pat10), Patient 22 (Pat22), Patient 25 (Pat25), and Patient 26 (Pat26); and three with the same expression profile, Patient 8 (Pat8), Patient 16 (Pat16), and Patient 18 (Pat18) (Appendix A). Three ATT (Pat9, Pat21, and Pat25) and two lung control (LC) samples (LC1 and LC2) were used as controls for comparisons (Appendix A). Because *ITF2* was mainly downregulated in NSCLC patients, while *DKK1* showed a more heterogeneous expression pattern, we considered *DKK1* as the best parameter to decide the bioinformatics analysis of the RNA-seq. The bioinformatic contrasts focused on three differential gene expression analyses: contrast A, tumor versus control; contrast B, comparison between tumors with high and low expression of *DKK1*; and contrast C, comparison between tumor samples with high expression of *DKK1* and controls (Figure 5). We selected the genes that showed significant expression differences (FDR < 0.05) in at least two of the three contrasts, prioritizing contrast B (Appendix A). The bioinformatics analysis also focused on all annotated genes related to the Wnt-pathway.

We analyzed the expression of nine candidates, but only four of them were confirmed by qPCR-PCR, three coding genes, *HOXD9*, *RIOX1,* and *CLDN6* and one long noncoding RNA, *XIST*. An accurate correlation with the RNA-sequencing data was found for *HOXD9, CLDN6*, and *XIST* genes (*r* = 0.83, *r* = 0.97 and *r* = 0.97, respectively) (Appendix A), while for *RIOX1* the correlation coefficient was less marked, probably because of the sample size (*r* = 0.58) (Appendix A). From all four candidate genes, only *HOXD9* expression showed a correlation with *ITF2* expression (Pearson = −0.24) (Figure 6A). In order to gain insight into the role of *ITF2* regulating the expression of *HOXD9* in NSCLC, we overexpressed *ITF2* in H23 resistant lung cancer cells. Transfection efficiency was confirmed by qRT-PCR at 24 and 72 h after transfection (Figure 6B). As expected from the primary tumors results, the overexpression of *ITF2* induced a significant decrease of *HOXD9* (*p* = 0.017) (Figure 6C). Having identified *HOXD9* as potential Wnt pathway candidate genes regulated by *ITF2*, we studied its clinical translational application in the cohort of NSCLC patients used for the RNA-seq analysis and in the public databases TCGA and KMplotter.

A negative correlation for *HOXD9* was found in terms of patients’ overall survival (Figure 6D). In addition, the survival analysis performed after stratifying patients according to the median of *HOXD9* gene expression showed that patients with lower expression of *HOXD9* presented a significantly better overall survival rate (*p* = 0.046) (Figure 6E). These results were confirmed when analyzing the expression levels of 1,926 lung cancer patients by using the Kaplan–Meier plotter online tool and the TCGA data sets, revealing that those patients with lower expression of *HOXD9* had a significantly better overall survival rate that was *independently* associated with the tumor histology (*p* < 0.001) (Figure 6F–H).

## 3. Discussion

Wnt signaling has recently been reported to be involved in driving platinum resistance of several tumor types [13,15]. However, the molecular mechanisms implicated are not clear, especially in NSCLC. In the present work, we have studied the involvement of the Wnt signaling pathway in tumorigenesis through a combined experimental approach by using both CGH arrays and RNA-sequencing. We have found that long term exposure to platinum induces a frequent deletion of *ITF2* that is involved, at least in part, in the activation of the Wnt pathway.

First, we identified a common deletion in H23, OVCAR3, and A2780 cells induced by cisplatin treatment in chromosome 18, including two completely deleted genes, *LRP1B*, and *ITF2.* A similar deletion was identified in a previous study analyzing cisplatin response in ovarian cancer samples, which supports our results [16]. However, there is another study analyzing the cytogenetic alterations of CDDP-resistant A2780R cells, which shows different genomic alterations, probably due to the specificity of the CGH-array used and the experimental design that included less representative number of probes and was performed in only one cell line [8]. We were not able to validate the *LRP1B* expression changes by using an alternative technique, an issue that has been previously reported [17]. In our case, it could be due to the mosaicism observed in this region that occurs in less than 20% of the resistant cells. The level of mosaicism that can be detected is dependent on the sensitivity and spatial resolution of the clones and rearrangements present [18]. Nevertheless, *LRP1B* could still play a role in tumor progression as several studies link its downregulation through deletion and carcinogenesis [19,20,21]. *ITF2* expression changes were confirmed in H23R and A2780R but not in OVCAR3R cells, also probably due to the level of mosaicism (36%) observed in these cells. Our results indicate that low levels of mosaicism would make the validations of expression changes by another quantitative technique difficult, probably because the alterations at expression levels are not significant enough to be detected.

The fact that *ITF2* is deleted and downregulated after platinum treatment provides us with new insight regarding its importance in resistance to platinum chemotherapy in lung and ovarian cancer. In line with our results, a previous study, using targeted sequencing in a PDX-based modeling of breast cancer chemoresistance, identified a genomic variant of ITF2 that depicted a link between its altered expression and breast cancer chemoresistance, although no detailed mechanism was provided to connect ITF2 function to chemoresistance [22].

ITF2 is a transcription factor belonging to the basic helix loop helix (bHLH) family, which can act as a transcriptional activator or repressor [23,24], but much about its regulation remains unknown. It is important to distinguish *ITF2*, whose official name is TCF4, from the T-Cell Factor 4 (TCF7L2), also known as *TCF4,* which is the ß-catenin transcriptional partner [25]. In fact, *ITF2* expression is induced by the ß-catenin/TCF complex, but at the same time, it acts as a repressor of this complex by interfering with the binding of ß-catenin to TCF4; this causes a decrease in the expression of Wnt target genes, leading to the repression of cell proliferation [12]. Consistent with these studies, we have observed that the resistant A2780 cells have increased activity of the ß-catenin/TCF transcription, which is concomitant with the increased expression of the downstream effector gene DKK1, probably due to the absence of ITF2. In contrast, we did not observe differences in H23 cells. However, a constitutive activity of ß-catenin/TCF was observed, a circumstance that was in agreement with previous reports which described that the H23 cells had an increased expression of Wnt2, which caused Wnt/bcat activation in an autocrine manner [26], and therefore, exogenous activation may not show a difference. Moreover, we have observed that the overexpression of *ITF2* in A2780R cells leads to a decrease in cell viability, rescuing the sensitive phenotype potentially through the inhibition of the excessive proliferation and the activity levels of the ß-catenin /TCF transcription. In fact, our results indicate that resistant cells respond better to the activation of the Wnt pathway, an effect that was restored after the re-expression of *ITF2*. Therefore, ITF2 plays an important role in the resistance to cisplatin probably through the regulation of the Wnt signaling pathway.

Our translational analysis based on the expression levels of *ITF2* and *DKK1* genes in two different cohorts of patients was aimed to elucidate the role of this pathway in tumor progression and chemotherapy response. *ITF2* expression was frequently downregulated in the NSCLC and ovarian tumor samples, validating our in vitro data. The expression levels of *DKK1*, however, showed a more heterogeneous pattern in the NSCLC tumor samples, while no differences were observed in the ovarian tumors, suggesting an aberrant activation of the Wnt signaling pathway in lung cancer. In fact, our in silico analysis of 1926 NSCLC patients indicates a significantly increased overall survival associated with high expression levels of *ITF2* and low expression of *DKK1.* The same findings, without statistical significance, were observed from our “in house” cohorts, probably due to the sample size. However, one of the strengths of our cohort was that it was comprised of fresh frozen samples, enabling us to perform high-quality RNA-sequencing in a group of NSCLC patients in order to determine the involvement of the Wnt-pathway components in lung cancer development. The differential expression of *DKK1* within the tumor samples allowed us to perform three different bioinformatics contrasts in order to explore all the possibilities regarding tumor development and the Wnt signaling pathway. Contrast A identified genes with a possible involvement in lung cancer development by comparing differential expression in tumors versus the control samples. Contrast B identified alterations in the Wnt pathway in NSCLC tumors and those that could be used as potential therapeutic targets. Finally, contrast C identified genes regulated by the Wnt pathway and others involved in NSCLC development. Using this approach, we were able to identify coding genes, noncoding genes, and transcripts that had not been functionally characterized previously [27,28]. Indeed, in this study, we have identified three coding genes, *HOXD9*, *CLDN6,* and *RIOX1*, and one noncoding gene, *XIST*, which could be involved in NSCLC progression through the Wnt signaling pathway. Our data was validated by two alternative methodologies (quantitative real-time and NGS), both showing strong positive correlations. From these candidates, we observed a negative correlation between *HOXD9* and *ITF2* expression levels. *HOXD9* was significantly downregulated in tumors with high expression levels of DKK1 and upregulated in tumors compared to controls, indicating that it could be involved in tumor progression through aberrant activation of the Wnt signaling pathway. We also observed that the tumor samples had higher expression levels of *HOXD9* than the controls, as it has been previously reported [29]. In addition, patients with lower levels of HOXD9 had better overall survival than those with upregulated expression of this gene. These results are consistent with previous studies linking a high expression of *HOXD9* with glioblastoma and hepatocarcinoma [30,31].

Additional functional analysis showed that *ITF2* overexpression in lung cancer cells H23R decreased the expression of *HOXD9*. Taking into account the basal activation of the Wnt pathway in this cell line, we believe that an alternative regulatory mechanism affected by *ITF2* is modulating the expression of *HOXD9*. Reinforcing this hypothesis, HOXD9 expression has been shown also to be regulated by epigenetic mechanisms such as the long noncoding RNA *HOTAIR* [32,33], which has been linked to cisplatin-resistance [34]. 

## 4. Materials and Methods

### 4.1. Cell Culture and Cell-Viability Assays

The NSCLC and ovarian cancer cell lines H23, H460, OVCAR3, and A2780 were purchased from the ATCC (Manassas, Virginia, USA) and ECACC (Sigma–Aldrich, Madrid, Spain) and cultured as recommended. Their CDDP-resistant variants H23R, H460R, OVCAR3R, and A2780R, were previously established in our laboratory [10,35]. Cisplatin (Farma Ferrer, Barcelona, Spain) was used for CDDP-viability assays. Cells were seeded in 24-well dishes at 40,000 cells/well, treated with increasing doses of CDDP (0, 0.5, 1, 1.5, 2, and 3 µg/mL) for an additional 72 or 48 h and stained as described [36]. Cell viability comparing sensitive vs. resistant cell lines was estimated relative to the density recorded over the same experimental group without drug exposure at the same period of time. Cell authentication is included in Appendix A.

### 4.2. Clinical Sample and Data Collection

We selected a representative number of fresh frozen surgical specimens from University Hospital La Paz (HULP)-Hospital Biobank, totaling 25 NSCLC and 10 ovarian cancer samples, belonging to previously reported cohorts of patients [10]. Ten adjacent normal tissue (ATT) samples from NSCLC patients, two additional lung tissue samples of non-neoplastic origin from autopsies and ten more normal ovarian samples obtained from sex reassignment surgery or tubal ligation were used as negative controls (NC). All tumor patients had both a perioperative PET-CT scan showing localized disease and pathological confirmation of stages after having undergone a complete resection for a histologically confirmed tumor. The samples were processed following the standard operating procedures with the appropriate approval of the Human Research Ethics Committee at IdiPAZ (Ethic code: PI-3508), including informed consent within the context of research. Clinical follow-up was conducted according to the criteria of the medical oncology division, pathological, and therapeutic data were recorded by an independent observer, and a blind statistical analysis was performed on these data.

### 4.3. DNA Extraction and Array of Comparative Genome Hybridization

DNA from cell lines was isolated as previously described [37] and used to analyze copy number variations by the Array-CGH SurePrint G3 human CGH microarray 1 × 1 M (Agilent, Santa Clara, California, USA). Array experiments were performed as recommended by the manufacturer, described in detail in the GEO repository number GSE129692. The aberration detection method 2 (ADM-2) quality weighted interval score algorithm identified aberrant intervals in samples that had consistently high or low log-ratios based on their statistical score. The score represents the deviation of the weighted average of the normalized log-ratios from its expected value of zero calculated with the derivative Log2 ratio standard deviation algorithm. A fuzzy zero algorithm was applied to incorporate quality information about each probe measurement. Our threshold settings for the CGH analytics software to make a positive call were 6.0 for sensitivity, 0.45 for minimum absolute average log ratio per region, and 5 consecutive probes with the same polarity were required for the minimum number of probes per region.

### 4.4. RNA Extraction, RT-PCR, qRT-PCR

Total RNA from human cancer cell lines and surgical samples was isolated, reverse transcribed, and quantitative RT-PCR analysis was performed as previously described [10]. For RT-PCR, 2 µL of the RT product (diluted 1:5) was used for semi-quantitative PCR or qPCR reactions with Promega PCR Mix (Promega, Madison, Wisconsin, USA) and SYBR Green PCR Mix (Applied Biosystems, Waltham, Massachusetts, USA), respectively. RT-PCR was performed under the following conditions: (a) 1 cycle of 95 °C for 2 min; (b) 30–40 cycles of 95 °C for 1 min, 56–60 °C for 1 min, 72 °C for 1 min; (c) an extension of 5 min at 72 °C. qRT-PCR absolute quantification was calculated according to the 2^−∆Ct^ method using GAPDH as endogenous control, whereas relative quantification was calculated with the 2^−∆∆Ct^ using *GAPDH* as endogenous control and the sensitive-parental cell line as a calibrator. Samples were analyzed in triplicate using the HT7900 real-time PCR system (Applied Biosystems, USA). Primers and probes for qRT-PCR expression analysis were purchased from Applied Biosystems (*TCF4*, Hs00162613_m1; *LRP1B*, Hs01069120_m1; *DKK1*, Hs00183740_m1; *GADPH*, Hs03929097_g1). Primers for *DKK1*, *HOXD9*, *CLDN6*, *XIST,* and *RIOX1* for RT-PCR assays were designed, when possible, to analyze the specific transcript that showed significant changes in the RNA-seq; all primers and specific amplification conditions are listed in Appendix A. 

### 4.5. NGS (RNA-seq) and Wnt Signaling Pathways Analysis

Total RNA from nine tumor tissues, three lung-adjacent normal tissue (ATT) from NSCLC samples and two tissue samples of non-neoplastic origin from autopsies were sent to Sistemas Genómicos Company (Valencia, Spain) for RNA-sequencing. Library samples were prepared and sequenced as recommended by the manufacturer (Illumina, San Diego, California, USA) described in detail in the GEO repository number GSE127559. The bioinformatics analysis was performed in the HULP. Reads were analyzed to quantify genes and isoforms through the RSEM-v1.2.3 methodology (RNA-seq by expectation maximization) [38] and using the hg19 versions as reference for annotation. The differential expression was carried out with edgeR, which estimates the common and individual dispersion (CMN and TGW, respectively) to obtain the variability of the data [39]. P-values and FDR statistical analyses were performed by Cmn and Twg models and the statistical cut-off point was set as FDR < 0.05. Normalization was performed by the TMM method (trimmed mean of M-values) [40]. The bioinformatics analysis included an efficiency analysis for every sample, considering the total efficiency as the percentage of reads annotated belonging to a transcript regarding the total fragments initially read. When using principal component analysis (PCA), no differences were observed between samples from non-neoplastic autopsies and adjacent normal tissue (ATT) from NSCLC patients in terms of transcriptomic profile, therefore both types of samples were considered as reference groups for the differential expression analysis (Appendix A). Three different bioinformatics contrasts are described in detail in the results section. 

### 4.6. Transfection Assays: Top-fop, B-cat and TCF4 Overexpression

A Myc-DDK-tagged ORF clone of *TCF4* and the negative control pCMV6 were used for in transient transfection (RC224345; OriGene, Rockville, MD, USA) using previously described methodology [10]. Cells were plated onto 60 mm dishes at 600,000 cells/dish and transfected with a negative control or *TCF4* vectors, using jet-PEI DNA transfection reagent (PolyPlus Transfection, Illkirch, France). For the Wnt reporter, assay cells were plated at a concentration of 600,000 cells/well in 6 MW plates. Cells were serum-starved overnight and cotransfected with 0.2 μg of either Super8xTopFlash (containing 7 copies of the TCF/LEF binding site) or Super8xFopFlash (containing 6 mutates copies of the TCF/LEF binding sites) expression plasmids, and 0.1 μg pRL-TK (Renilla-TK-luciferase vector, Promega, Madison, WI, USA) as a control, using lipofectamine 2000. Cells were subsequently treated with LiCl 10 mM or cotransfected with S33Y ß-catenin for 48 h prior to luciferase activities being measured using a Glomax 96 Microplate Luminometer (Turner Biosystems Instrument, Sunnyvale, CA, USA). Firefly luciferase activity was calculated as light units normalized with the Renilla activity generated by the pRL-CMV vector. ß-catenin/TCF activity was calculated by obtaining the ratio of the Top/Fop promoter activities and expressed in relative terms as the fold change of the untreated cells activation levels (=1).

### 4.7. In Silico Databases: The Cancer Genome Atlas and Kaplan–Meier Plotter

The Cancer Genome Atlas (TCGA) data: We selected samples from the TCGA data sets containing RNA sequencing information: 230 lung adenocarcinoma, 487 lung SCC, 186 esophageal adenocarcinoma and 522 head and neck SCC tumors, to analyze the expression of *ITF2* and *HOXD9* genes. We obtained overall survival Kaplan–Meier estimations selecting groups of patients with high and low expression of *ITF2/HOXD9* (mRNA expression Z-score threshold = 1). Samples with *ITF2* deletion and low expression were included in the *ITF2* low group.

Kaplan–Meier Plotter: We obtained the survival analysis of 1926 patients from the Kaplan–Meier Plotter online tool [41] for the Affymetrix ID probes 213891_s_at for *ITF2*, 204602_at for *DKK1* and 206604_at for *HOXD9.* Groups were separated according to the median in high and low expression.

### 4.8. Statistical Analysis

Data were compared using the chi-squared test or Fisher’s exact test for qualitative variables, and Student’s *t*-test or the Wilcoxon–Mann–Whitney test for quantitative variables. The correlation of quantitative variables was analyzed by Pearson’s test. Overall survival was estimated according to the Kaplan–Meier method and compared between groups by means of the log-rank test. All the p-values were two-sided, and the type I error was set at 5 percent. Statistical analyses were performed using SPSS_20 software (IBM, Armonk, NY, USA).

## 5. Conclusions

In essence, we have identified *ITF2* as a frequently downregulated gene in cisplatin-resistant cancer cells as well as in NSCLC and ovarian cancer patients. We have also observed a statistically significant relationship with a better response to platinum treatment not only in the lung but also in other epithelial tumors, suggesting that *ITF2* could be used as a general epithelial tumor platinum-predictive marker. Moreover, we have defined its implication as a molecular mechanism behind the development of cisplatin resistance in cancer cells probably through the activation of the Wnt-signaling pathway and, several of its downstream effector genes, providing novel insights into the molecular biology and the cellular mechanisms involved in the acquired resistance to the most widely-used chemotherapy agent, cisplatin. Finally, we have suggested two potential therapeutic targets for further study, *ITF2* and *HOXD9.*

## Figures and Tables

**Figure 1 cancers-12-00786-f001:**
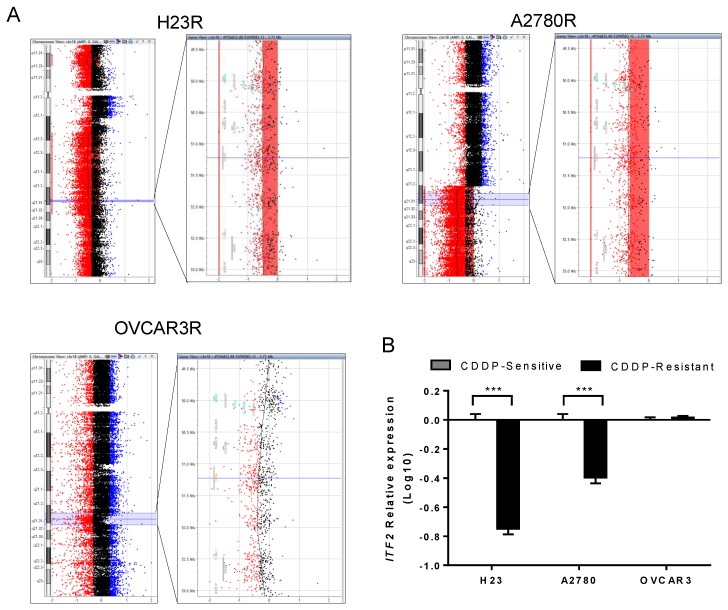
Identification of a common deletion in chromosome 18 in cisplatin-resistant cancer cell lines. (**A**) Picture extracted from the Agilent cytogenomics 3.0.1.1 software showing the *ITF2* deletion in chromosome 18 in H23R, A2780R, and OVCAR3R cell lines. (**B**) Relative mRNA expression levels of *ITF2* measured by qRT-PCR. The results show the mean fold induction compared to the sensitive cells. Gene expression was normalized to *GAPDH*. S, sensitive; R, resistant; data represents the relative expression levels obtained from the combination of two independent experiments measured in triplicate ± SD. *** *p* < 0.001; (Students *t*-test).

**Figure 2 cancers-12-00786-f002:**
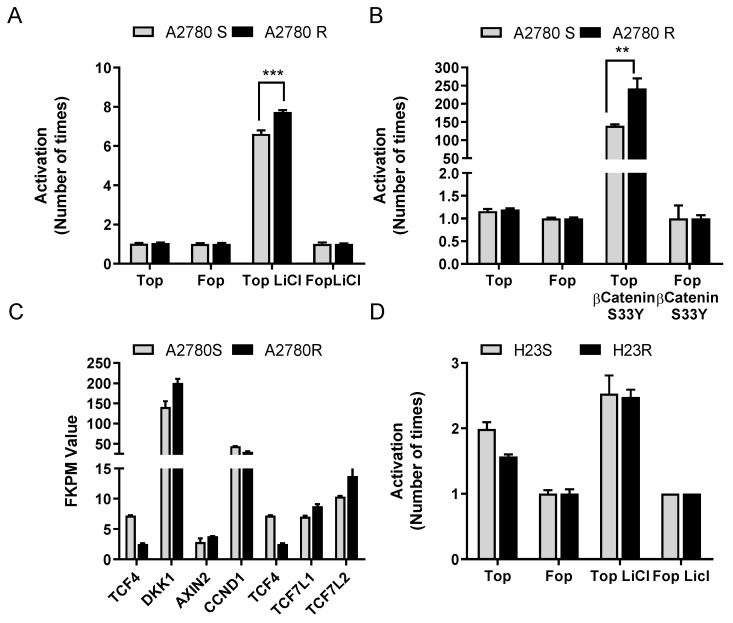
The basal status of the Wnt signaling pathway in A2780 and H23 cell lines. (**A**–**B**) Pharmacological activation (A) and functional activation (B) of β-catenin transcriptional activity in A2780 cells. (**C**) The expression levels of downstream genes regulated by *ITF2* involved in Wnt signaling pathway in A2780S and A2780R cells measured by RNAseq in terms of “fragments per kilobase of transcript per million mapped reads” or FPKM values. (**D**) Pharmacological activation of β-catenin transcriptional activity in H23 cells. β-catenin transcriptional activity was measure in A2780 and H23 cells after treatment with LiCl (10 mM) 24 h or transfection of bcat-S33Y, transfecting with Super8xTopFlash (Top) or Super8xFopFlash (Fop). The results show the fold induction of the Top/Fop ratio with respect to untreated cells (= 1). Values represent the mean of three independent experiments measured by triplicate ± SD. *** *p* < 0.001; ** *p* < 0.01, **p* < 0.05 (Students *t*-test) ns, non significant.

**Figure 3 cancers-12-00786-f003:**
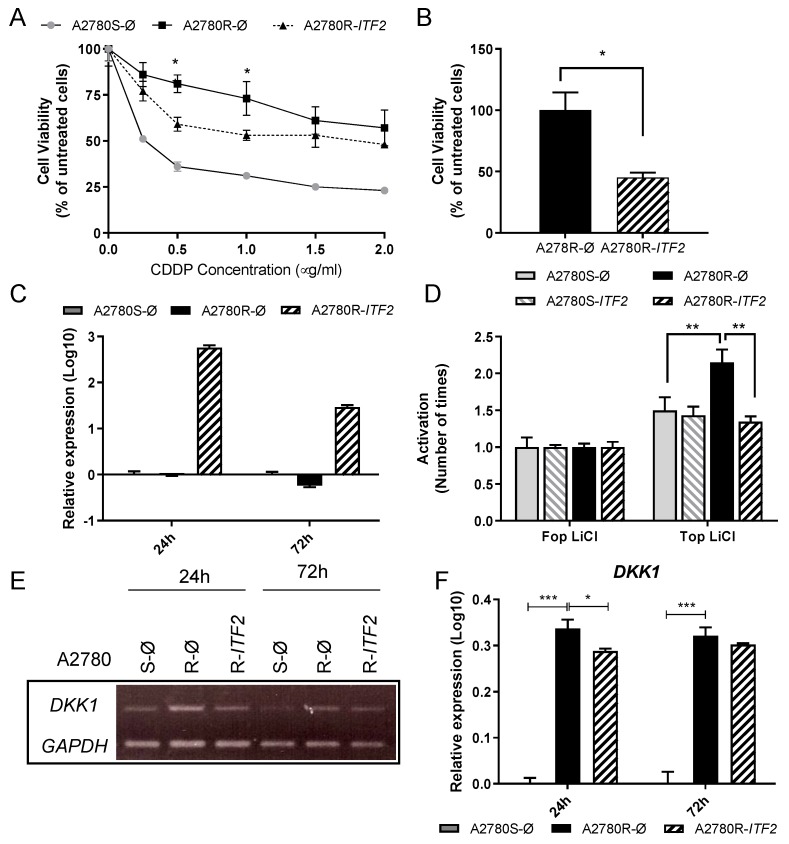
The effect of *ITF2* on cisplatin resistance, cell viability, and Wnt pathway. (**A**) Viability curves of A2780 cell lines transfected with pCMV6 (S-Ø and R-Ø) and with the overexpression vector (R-ITF2). Each experimental group was exposed to six different CDDP concentrations for 48 h. Data were normalized to each untreated control, set to 100%. The data represent the mean ± SD of at least three independent experiments performed in quadruplicate at each drug concentration for each cell line analyzed. (**B**) Viability of A2780 cell lines transfected with pCMV6 (R-Ø) and with the overexpression vectors (R-ITF2). (**C**) Relative expression levels of *ITF2* measured by quantitative RT-PCR represented in Log10 scale; in each experimental group, the sensitive cell line transfected with pCMV6 plasmid was used as a calibrator. Each bar represents the combined relative expression of two independent experiments measured in triplicate. (**D**) β-catenin transcriptional activity was measured in A2780 cells after *ITF2* overexpression and treatment with LiCl (10 mM) for 24 h, transfected with Super8xTopFlash (Top) or Super8xFopFlash (Fop). The results show the fold induction of the Top/Fop ratio with respect to untreated cells (=1). Values represent the mean of three independent experiments measured by triplicate ± SD. (**E**) Expression analysis of the downstream gene DKK1 regulated by *ITF2* in A2780 cell line transfected with pCMV6 (S-Ø and R-Ø) and with the overexpression vector (R-ITF2) for 24 and 72 h. Representative images of *DKK1* and *GAPDH* measured by RT-PCR. (**F**) Expression levels of *DKK1* measured by qRT-PCR. Each assay was performed at least three times to confirm the results. *** *p* < 0.001; * *p* < 0.05 (Students *t*-test).

**Figure 4 cancers-12-00786-f004:**
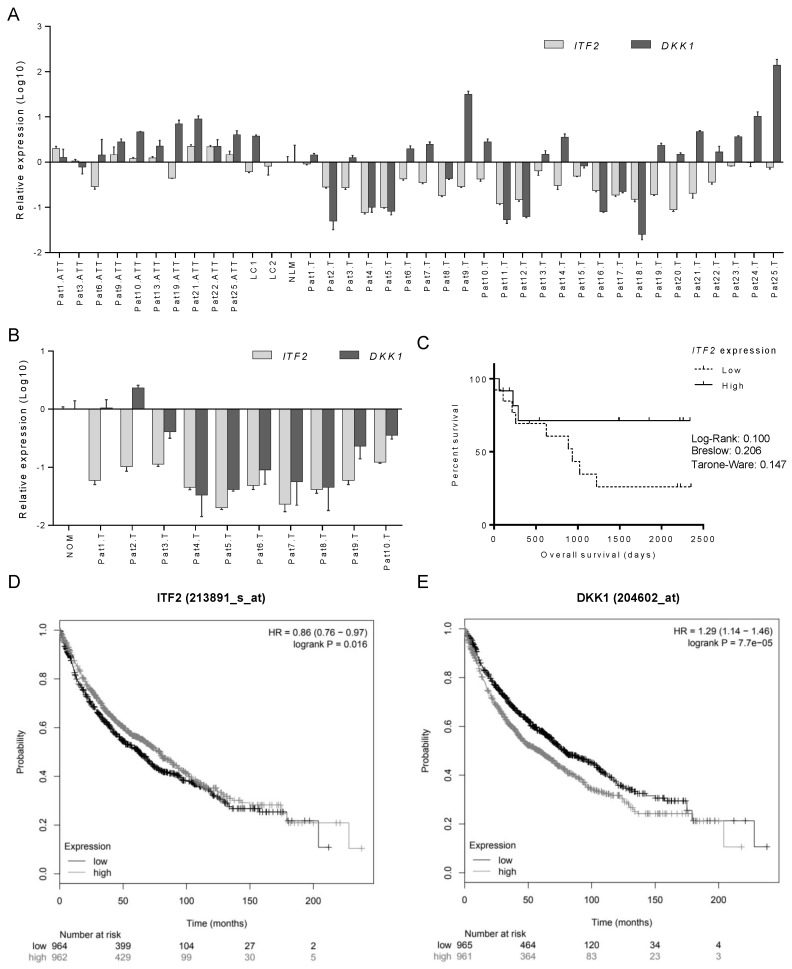
The expression profile of *ITF2* and *DKK1* in patients with NSCLC and ovarian cancer. (**A**,**B**) Assessment of *ITF2* and *DKK1* expression levels measured by qRT-PCR in 55 fresh samples from nontumor samples, and two cohort NSCLC (**A**) and ovarian cancer patients. (**B**) For all the analyses, data represents expression levels in 2−∆ΔCt using the mean of normal lungs (NLM) or ovarian (NOM) as calibrator. (**C**) Survival analysis in 25 NSCLC samples according to the mean of *ITF2* expression. LogRank, Breslow, and Tarone–Ware tests were used for comparisons, and *p* < 0.05 was considered a significant change in OS. NSCLC, non-small cell lung cancer; ATT, adjacent tumor tissue; T, tumor; LC1/LC2, Lung Control; NLM, normal lung mean; NOM, normal ovarian mean. (**D**,**E**) Survival analysis in 1926 patients from the Kaplan–Meier online tool for ITF2 (**D**) and DKK1 (**E**) in terms of overall survival.

**Figure 5 cancers-12-00786-f005:**
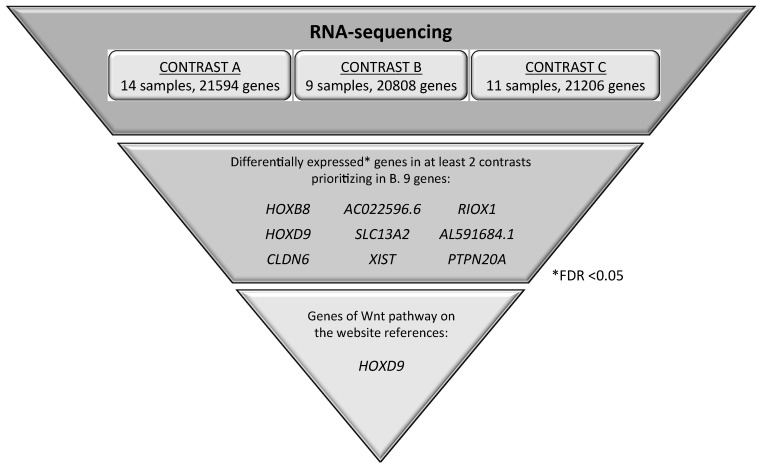
Selected genes potentially involved in the Wnt signaling pathway. Genes were identified through a global transcriptomic analysis of 14 NSCLC patient samples combining the information with all annotated genes related to the Wnt-pathway in gene sets of the molecular signatures database (MSigDB, Broad Institute).

**Figure 6 cancers-12-00786-f006:**
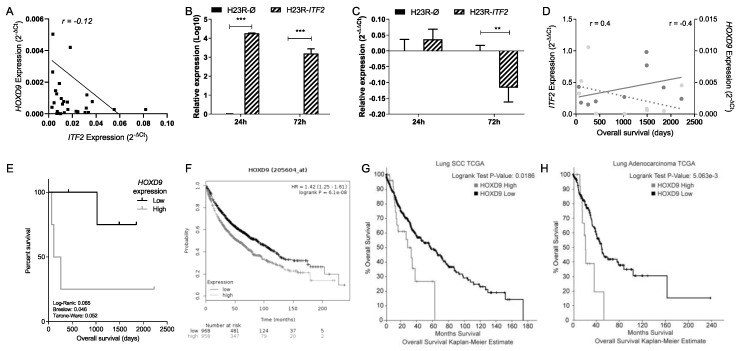
Analysis and clinical significance of *ITF2* and *HOXD9* in NSCLC samples. (**A**) Correlation between *ITF2* expression and *HOXD9* in tumor and nontumor samples from the complete cohort of 25 NSCLC patients. The Pearson coefficient was used for the linear correlation of the quantitative variables. (**B**,**C**) Effect of *ITF2* overexpression on *HOXD9* levels (B) Validation of the transfection efficiency of *ITF2* at mRNA levels. Relative expression levels of *ITF2* measured by qRT-PCR, in the cell line H23R, at 24 and 72 h after transfection represented in Log10 the 2^−ΔΔCt^. (C) Relative expression levels of *HOXD9* measured by quantitative RT-PCR after *ITF2* overexpression. For both (B) and (C) the resistant cell line transfected with pCMV6 plasmid was used as a calibrator (R-Ø). H23R cells were also transfected with ITF2 cDNA (R-ITF2). Each bar represents the combined relative expression of two independent experiments measured in triplicate. *** *p* < 0.001; ** *p* < 0.01 (Students *t*-test). (**D**) Correlation between *ITF2* and *HOXD9* expression levels with the overall survival analyzed in NSCLC patients selected form the RNA-seq analysis. The left Y axis represents the quantitative expression levels of ITF2 (grey circles) measured by qRT-PCR as 2^−∆∆Ct^ and its correlation with overall survival in days (grey line, *r* = 0.4). The right Y axis indicates the quantitative expression levels of *HOXD9* (black triangles) measured by qRT-PCR as 2^−∆Ct^ and its correlation with overall survival in days (black dotted-line, *r* = −0.4). (**E**) Survival analysis in NSCLC samples according to the mean of *HOXD9*. LogRank, Breslow, and Tarone–Ware tests were used for comparisons and *p* < 0.05 was considered as a significant change in overall survival (OS). (**F**–**H**) Survival analysis in 1926 patients from the Kaplan–Meier online tool, (**F**) and the TCGA data sets of lung adenocarcinoma, (**G**) high = 23; no change = 203; and SCC, (**H)** high = 28; no change = 441 for *HOXD9*.

**Table 1 cancers-12-00786-t001:** Clinicopathological and experimental data obtained from patients with NSCLC from La Paz University Hospital.

Patient	Histology	Sex	Stage	Chemotherapy	Relapse	Status	OS (days)	PFS (days)	*TCF4*	*DKK1*	*HOXD9*	*RIOX1*
									(2^−ΔCt^)	(2^−ΔCt^)	(2^−ΔCt^)	(2^−ΔCt^)
Pat1.T	Adenocarcinoma	Female	IA	No	Yes	Alive	2220	1490	0.03364	0.00089	0.00008	0.00872
Pat2.T	Adenocarcinoma	Male	NA	No	Yes	Alive	2352	1860	0.01041	0.00003	0.04277	0.01605
Pat3.T	Epidermoid	Male	IB	No	Yes	Exitus	1022	825	0.01012	0.00075	0.00001	0.00114
Pat4.T	Adenocarcinoma	Male	IB	No	No	Exitus	3	3	0.00286	0.00006	0.00041	0.00624
Pat5.T	Adenocarcinoma	Male	NA	No	No	Exitus	626	626	0.00365	0.00005	0.00503	0.00932
Pat6.T	Large cell	Male	IIB	No	No	Exitus	62	62	0.01584	0.00119	0.00074	0.00216
Pat7.T	Adenocarcinoma	Male	IIIA	Otro	Yes	Exitus	228	138	0.01278	0.00150	0.00078	0.12070
Pat8.T	Epidermoid	Female	IIIB	CDDP + Others	No	Exitus	109	109	0.00675	0.00026	0.00126	0.00196
Pat9.T	Adenocarcinoma	Female	IIA	CDDP + Others	Yes	Alive	2260	2260	0.01070	0.01826	0.00097	0.04697
Pat10.T	Epidermoid	Male	IB	No	No	Alive	1853	1853	0.01626	0.00162	0.00027	0.00723
Pat11.T	Adenocarcinoma	Male	IA	No	No	Exitus	216	216	0.00456	0.00003	0.00070	0.01399
Pat12.T	Adenocarcinoma	Female	IIIA	CBDCA + Others	Yes	Alive	2192	2192	0.00580	0.00004	0.00146	0.01059
Pat13.T	Epidermoid	Male	IB	CDDP + Others	No	Alive	2341	2341	0.02466	0.00090	0.00013	0.14752
Pat14.T	Epidermoid	Male	IIA	No	ND	Exitus	289	289	0.01179	0.00206	0.00061	0.02547
Pat15.T	Epidermoid	Male	IIA	No	ND	ND	109	109	0.01833	0.01370	0.00420	0.07619
Pat16.T	Adenocarcinoma	Female	IIIA	CDDP + Others	ND	Alive	2228	2228	0.00907	0.00005	0.00191	0.00814
Pat17.T	Adenocarcinoma	Male	IIB	Otro	Yes	Exitus	888	443	0.00704	0.00014	0.00002	0.00833
Pat18.T	Epidermoid	Male	IIB	CBDCA + Others	No	Exitus	259	259	0.00563	0.00001		0.01455
Pat19.T	Adenocarcinoma	Female	IB	CDDP + Others	Yes	Exitus	936	428	0.00738	0.00137	0.00014	0.01572
Pat20.T	Epidermoid	Male	IIB	CDDP + Others	Yes	Exitus	1224	637	0.00346	0.00089	0.00264	0.00422
Pat21.T	Adenocarcinoma	Male	IIIA	CDDP + Others	No	ND	421	421	0.00792	0.00279	0.00003	0.00509
Pat22.T	Adenocarcinoma	Female	IIB	CDDP + Others	ND	ND	184	184	0.01319	0.00101	0.00007	0.01480
Pat23.T	Epidermoid	Male	IIIA	CDDP + Others	ND	ND	542	542	0.03221	0.00214	0.00120	0.01741
Pat24.T	Adenocarcinoma	Male	IIA	CBDCA + Others	No	Alive	1491	1491	0.03680	0.00614	0.00077	0.01088
Pat25.T	Adenocarcinoma	Male	IIA	CDDP + Others	No	Alive	1496	1496			0.00008	0.00872

Note: OS, Overall Survival; PFS, progression free survival; CDDP, cisplatin; CBDCA, carboplatin; NA, not available.

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
