# Peer review of "A Novel Role for the Tumor Suppressor Gene ITF2 in Tumorigenesis and Chemotherapy Response"

_cancers, 2020, doi:10.3390/cancers12040786_

Round 1

Reviewer 1 Report

Authors present a very well conceived study on potential mechanism of cisplatin resistance in NSCLC and ovarian cancer. Experiments are solid and results are robust. Paper is well written. Conclusions are supported by results and they are presented in an appropriate way.

Author Response

Nothing to add except our gratitude for the critical analysis and positive evaluation of our manuscript

Reviewer 2 Report

In this study, the authors identified the deletion of ITF2 in cisplatin-resistant H23 human NSCLC cells and A2780R and OVCAR3R human ovarian cancer cells and ovarian cancer cells lung tumorigenesis, presented the expression profile of ITF2 in patients with NSCLC and ovarian cancer, and demonstrated that ITF2 loss was associated with a worse overall survival in patients with lung cancer esophageal adenocarcinoma, or head and neck SCC. The authors concluded that deletion of ITF2 is involved in the development of cisplatin resistance probably through the activation of the Wnt-signaling pathway, suggesting the possible role of ITF2 as a novel therapeutic target for platinum resistant tumors.

  1. The title of the manuscript, which indicated that the present study was on the role of ITF2 in lung tumorigenesis and chemotherapy response, did not reflect the actual work conducted in the present study. In this study, the authors have examined the association of ITF2 with cisplatin resistance in both human lung and ovarian tumor cell lines and the expression of ITF2 in both human lung and ovarian tumor tissue samples. What was the reason that ovarian cancer was not mentioned in the title?
  2. Line 69 – 70: “We found two deleted regions shared by both 69 tumor types in at least three of the four cell lines (H23R, A2780R and OVCAR3R)…” What is the common oncogenic driver mutation in those three cancer cell lines that is not present in H460?
  3. Line 120 – 121: “In order to test the role of ITF2 in cisplatin resistance through the Wnt pathway, we transiently 120 overexpressed ITF2 cDNA in A2780 cells”. Why was A2780 ovarian cancer cells used but not H23 NSCLC cells if the study was focused on lung cancer as indicated in the title of the manuscript?
  4. As mentioned in Line 342, “Cells were seeded in 24-well dishes at 40,000 cells/well, treated with 342 increasing doses of CDDP (0, 0.5, 1, 1.5, 2 and 3µg/ml) for an additional 72 or 48 hours”. Figure 3A shows the cell viability of three A2780 subclones after 48hr CDDP treatment. What were the results for 72 hr treatment and other cancer cell lines (i.e., H23, H460 and OVCAR3)? The authors should increase the CDDP concentration range so that the IC50 values could be calculated for individual cells lines. The IC50 value is a better indicator of the relative drug sensitivity in difference cell lines than the cell viability value at a particular drug concentration.
  5. Figure 3B. What was the treatment conditions for Fig. 3B?
  6. In Line 104 - 105, it was indicated that the significant decrease of ITF2 expression in the cisplatin resistant phenotype was associated with an increase in DKK1. However, in Line 215, it was stated that “From all four candidates, only HOXD9 expression showed correlation with ITF2 expression.” What was the DKK1 expression status in those four candidates? Was DKK1 correlated with HOXD9 in those four candidates? As the authors concluded in Line 33 and 34, “this data highlights the possible role of ITF2 and HOXD9 as novel therapeutic targets for platinum resistant tumors”. Although it was stated in Line 219 that “the overexpression of ITF2 induced a significant decrease of HOXD9” in ITF2-overexpressed CDDP-resistant H23 lung cancer cells, did the authors check if overexpression of ITF2 in CDDP-resistant H23 cells would restore their CDDP sensitivity like what they did with A2780 cells in Figure 3? If not, why would HOXD9 be a possible therapeutic target for platinum resistant tumors but not DKK1?
  7. Line 424-425, what was the rationale that the esophageal adenocarcinoma and head and neck SCC information but not the ovarian cancer information was included in the analysis?

Author Response

Answer to Referee 2

Point1: The title of the manuscript, which indicated that the present study was on the role of ITF2 in lung tumorigenesis and chemotherapy response, did not reflect the actual work conducted in the present study. In this study, the authors have examined the association of ITF2 with cisplatin resistance in both human lung and ovarian tumor cell lines and the expression of ITF2 in both human lung and ovarian tumor tissue samples. What was the reason that ovarian cancer was not mentioned in the title?

One of the translational strengths of our manuscript is the high RNA quality from fresh-frozen tumor samples, mainly from patients harboring early stages of non-small cell lung cancer, which allowed us to perform a comprehensive transcriptome study to delve into the potential role of ITF2. For this reason we decided to emphasize the mention of lung cancer in the title. It was decided to include tumorigenesis in the title because the functional experiments of the Wnt pathway with TOP / FOP vectors, with and without ITF2 overexpression, demonstrate modifications of the Wnt pathway and do not imply treatment with platinum. In addition in most of tumors analyzed both, our cohorts and in the "in silico" interrogated cohorts, showed lower levels of ITF2 expression than in normal tissues, before any chemotherapy treatment.

However, we understand the reviewer's concern by including only one tumor type in the title when, throughout the manuscript, data related to other tumors of epithelial origin are presented. We therefore propose modifying the title of “A novel role for the tumor suppressor gene ITF2 in lung tumorigenesis and chemotherapy response”, by “A novel role for the tumor suppressor gene ITF2 in tumorigenesis and chemotherapy response”

Point 2: Line 69 – 70: “We found two deleted regions shared by both tumor types in at least three of the four cell lines (H23R, A2780R and OVCAR3R)…” What is the common oncogenic driver mutation in those three cancer cell lines that is not present in H460?

We found two common chromosomal regions deleted in the H23, A2780 and OVCAR3 cell lines after having established them resistant to cisplatin. These regions were 18q21.2-18q21.31 and 18q21.32. These deletions were not observed in the H460 cells. Since the initial objective of the manuscript was the evaluation of the effect of cisplatin, we focused on the study of the three cell lines with common cytogenetic alterations after treatment with the drug.  

Following the reviewer's consideration, we have assessed the mutations described in the literature for the 4 cell lines, which can be seen in the attached table. There is no common driver mutation between these three selected lines, nor different from the H460 cell line. Therefore, we understand that the frequent deletion observed in these regions of chromosome 18 must be associated with a greater susceptibility to cisplatin binding. In fact cisplatin kills cancer cells by damaging their DNA, mainly by forming Pt-d (GpG) and, to a lesser extent, Pt-d (ApG), Pt-d (GpXpG) intrastrand diadduct, and at lower frequency, Pt -GG interstrand cross-links (PNAS October 11, 2016 113 (41) 11507-11512; https://doi.org/10.1073/pnas.1614430113).

H23

H460

A2780

OVCAR3

The cells carry the K-ras 12 mutation, and there is a mutation in codon 246 (ATC -> ATG, isoleucine -> methionine) of the p53 gene.

The cells express easily detectable p53 mRNA at levels comparable to normal lung tissue, and exhibit no gross structural DNA abnormalities.

The cells carry SMARCA4, RRAS3  PIK3CA and PTEN  genes mutations

The cells carry TP53 and PIK3R1 genes mutations

There is expression of C-myc, L-myc, v-src, v-abl, v-erb B, c-raf 1, Ha-ras, Ki-ras and N-ras RNAs.

The cells stain positively for keratin and vimentin but are negative for neurofilament triplet protein.

 Several normal chromosomes (N11, N13, N14, N15, N16, N17, and N22) are clearly under-represented. Many of these missing chromosomes are represented in the large number of cytogenetically altered chromosomes identified as marker chromosomes.

The cells express heterogeneous mRNA expression for PDGF A and B chain, transforming growth factor alpha and beta and the epidermal growth factor receptor (EGFR).

In addition to the marker chromosomes, there are a large number of other structurally abnormal and unassignable chromosomes that are not recognized as markers.

NCI-H23 exhibits a high degree of c-myc DNA amplification (20-fold) but no detectable amplification of c-myc RNA.

Random loss and gain of chromosomes from cell to cell are noted in the exact chromosome counts and in the analysis of the karyotypes.

The cells stain positive for keratins 5+8 and 18 and vimentin but are negative for neurofilament.

Point 3: Line 120 – 121: “In order to test the role of ITF2 in cisplatin resistance through the Wnt pathway, we transiently overexpressed ITF2 cDNA in A2780 cells”. Why was A2780 ovarian cancer cells used but not H23 NSCLC cells if the study was focused on lung cancer as indicated in the title of the manuscript?

Our data show that platinum treatment, frequently induces the indicated deletion (in 3 of the four cell treated) and that it affects the entire ITF2 gene, while in the cell line H460 this deletion was not observed. Knowing the function of this gene in the Wnt pathway, we wanted to investigate how this pathway could be affected by the deletion ofITF2after treatment with platinum. However, it was impossible to assess changes in Wnt pathway in the H23 cell line because we observed a constitutive activity of bcat/TCF, circunstance that was in agreement with previous reports describing that the H23 cells have an increased expression of Wnt2, which causes Wnt/bcat activation in an autocrine manner. (Oncogene. 2009 May 28; 28(21): 2163–2172. doi:10.1038/onc.2009.82). This is why we consider that in H23 cells it will be hard to manipulate the pathway.

We have clarified this condition in the manuscript, lines 359-363 as follows: “In contrast, we did not observe differences in H23 cells, but a constitutive activity of bcat/TCF, circunstance that was in agreement with previous reports describing that the H23 cells have an increased expression of Wnt2, what causes Wnt/bcat activation in an autocrine manner [26]therefore, exogenous activation may not show a difference.”

Nevertheless our “in silico” analysis in 1.053 patients with NSCLC from TCGA databases identified a frequent loss of expression of the ITCF4/TF2 gene (8.65% in adenocarcinomas and 6,8% in squamous cell carcinoma) validating what we observed when comparing tumor and non-pulmonary pathologies in the fresh samples collected in our laboratory. Furthermore, we analyzed lung human patient samples and could confirm the predictive value of our findings in vitro, showing that TCF4 deletion andDKK1over-expression can predict the outcome of these patients. Taking into account all the compiled data, we could not rule out that the loss of TCF4/ITF2 is affecting the tumorigenic process in NSCLC patients only because we couldn’t assay the Wnt pathway in H23 cells. To look for stronger evidences of the function of the Wnt pathway, we analyzed the transcriptomeand the components of this pathway in primary tumors form NSCLC patients samples for which we had quality RNA as we have described in response number 1. We carried out the functional tests of the Wnt pathway on the A2780 cell line, as it is a clear model in which the situation of resistance to cisplatin was concomitant with a deletion of ITF2 and an activation of Wnt.

Point 4: As mentioned in Line 342, “Cells were seeded in 24-well dishes at 40,000 cells/well, treated with increasing doses of CDDP (0, 0.5, 1, 1.5, 2 and 3µg/ml) for an additional 72 or 48 hours”. Figure 3A shows the cell viability of three A2780 subclones after 48hr CDDP treatment. What were the results for 72 hr treatment and other cancer cell lines (i.e., H23, H460 and OVCAR3)? The authors should increase the CDDP concentration range so that the IC50 values could be calculated for individual cells lines. The IC50 value is a better indicator of the relative drug sensitivity in difference cell lines than the cell viability value at a particular drug concentration.

Indeed we appreciate the fact that the reviewer has noticed an error in the description of the experimentation times in this section. Actually, it should be stated that the cells were treated for 48 hours with cisplatin and the curves were stopped 72 hours after the "in transient" ITF2overexpression. Therefore, the results shown in Figure 3B correspond to 72h when the experiment is stopped.

Cisplatin viability curves after overexpressing ITF2in H23 cells were previously performed in our laboratory and are now introduced as Supplementary figure 2.As expected due to the constitutive activity of the Wnt pathway described in H23 cells, no changes were observed after transfection in H23 resistant cells.  Due to the inclusion of this new Supplementary figure 2, the old Supplementary Figures  2, 3 and 4, are now Figures 3, 4 and 5.

We have introduced these results in the body text in lines 127-132 as follows:  “In order to test the role of ITF2 in cisplatin resistance through the Wnt pathway, we transiently overexpressed ITF2 cDNA in A2780 and H23 cells. Cell sensitivity to Cisplatin was unmodified after ITF2 overexpression in H23 cells as expected, probably due to the constitutive activity of the Wnt pathway already described in this cell line (Supplementary Figure 2). However, our previous results indicate that A2780 cells seem to be a reliable cellular model to evaluate changes in the transcriptional activity of the Wnt pathway.

We did not performed this functional test on the H460 lines since this line was discarded as an object of study as the deletion that affected the ITF2 gene was not observed in these cells as explained in concern 3. Regarding the OVCAR3 cell line, the deletion of ITF2was observed with 36% mosaicism, an alteration that was not enough to detect a significant decrease in gene expression as shown in Figure 1B. This fact is discussed in lines 276-279 of the manuscript, and would rationalize and justify the fact that in the absence of a decrease in expression, gene transfection assays were not carried out in this cell line. However, we have considered providing the following data to the reviewer that supports our quantitative results in OVCAR3 cells. We are showing a comparison of TCF4 expression levels between sensitive and resistant cells. These data are extracted from the RNAseq data obtained in our cellular models. As we can see in the figure, the 30% of mosaicism observed in OVCAR3R cells does not correlate with a decrease in TCF4 expression.

Point 5: Figure 3B. What was the treatment conditions for Fig. 3B?
Answer included in Concern 4.

Point 6: In Line 104 - 105, it was indicated that the significant decrease of ITF2 expression in the cisplatin resistant phenotype was associated with an increase in DKK1. However, in Line 215, it was stated that “From all four candidates, only HOXD9 expression showed correlation with ITF2 expression.” What was the DKK1 expression status in those four candidates? Was DKK1 correlated with HOXD9 in those four candidates?. As the authors concluded in Line 33 and 34, “this data highlights the possible role of ITF2 and HOXD9 as novel therapeutic targets for platinum resistant tumors”. Although it was stated in Line 219 that “the overexpression of ITF2 induced a significant decrease of HOXD9” in ITF2-overexpressed CDDP-resistant H23 lung cancer cells, did the authors check if overexpression of ITF2in CDDP-resistant H23 cells would restore their CDDP sensitivity like what they did with A2780 cells in Figure 3? If not, why would HOXD9 be a possible therapeutic target for platinum resistant tumors but not DKK1?

These authors think that there is a misunderstanding between candidates and samples. We apologize for the confusion on the analysis. When we write “four candidates” we are referring to four candidate genes, no patients. 

As the reviewer mentions, DKK1 expression is inversely correlated with ITF2 in 56% of the samples from our cohort. Since ITF2 was low in all the NSCLC samples from our cohort, one of the analyses was done stratifying samples based on DKK1 expression as it presents more heterogeneous expression pattern, including samples with DKK1 low and high expression. The 4 candidate genes come from the differential expression analysis, and they all correlate with DKK1 high expression. In addition, both, DKK1 and Hoxd9 high expression, can predict worse patient outcome, indicating that both could be used as predictive and therapeutic targets.

We address the concerns regarding the H23 cell line in comment number 3

These authors consider it can be clarified if we change in the text “From all four candidates” by “from all four candidate genes” in line 263.

Point 7: Line 424-425, what was the rationale that the esophageal adenocarcinoma and head and neck SCC information but not the ovarian cancer information was included in the analysis?

In the light of our findings, and that several epithelial tumors share common oncogenic drivers and chemotherapy-resistant mechanism, we explored ifTCF4 deletion was present in other epithelial tumors and if could predict patient outcome. Head and neck SCC and esophageal carcinomas are both epithelial tumors that share expression signatures with lung squamous cancer, such asSOX2and TP63 amplifications.

We include the analysis of ovarian carcinomas available at TCGA. We could only identify 13 patients with deletions of TCF4or low expression, and 18 with an increased TCF4 expression out of 201, but there was no significant difference in overall survival between the two groups of patients. We hypothesize that in ovarian cancer, the presence of low TCF4 level could not predict overall survival, but as we demonstrated, it could explain cisplatin resistance. Since the TCGA samples are from patient non treated, we could just speculate about its function in drug resistance based in our in vitro data, and further analysis will need to be done using samples from treated patients, which are incredibly hard to obtain.

Reviewer 3 Report

The article thoroughly describes research using advanced molecular and bioinformatics tools. The content contains all the necessary information in the description of the methods and results. A large number of supplements is also required, including the STR profile of the tested lines. The tests were performed on cell lines, frozen clinical material and available in databases. Statistical analysis was performed using appropriate statistical tools. Appropriate controls were used for testests. The article can be published in its current form.

Author Response

(The authors gave the same response as above.)

Reviewer 4 Report

In the manuscript “A novel role for the tumor suppressor gene ITF2 in lung tumorigenesis and chemotherapy response” the athors identified 24 genes with a potential role in the development of lung and ovarian cancer. In paired platinum sensitive and resistant lung and ovarian carcinoma cell lines they found a common deletion that includes the Transcription Factor 4, TCF4, a downstream target gene of the Wnt/β-catenin pathway.

This manuscript has not a great novelty because the topic regarding Wnt/β-catenin pathway/TCF4 in lung cancer chemoresistance was previously addressed (i.e. doi:10.1186/s12885-015-1635-9; doi:10.1158/0008-5472.CAN-15-1804).

Overall the manuscript is confused in the starting hypothesis, the cellular models do not reflect the rationale, many experiments are shown in ovarian cancer cells and not in Lung cancer cells, KM analysis performed also in head and neck cancer patients using different datasets for each gene.

The authors had to choose a single pathology and focus on that, choosing a battery of cells representing different histological subgroups, in brief, fortifying and dissecting the chemoresistance mechanism that lacks in the manuscript.
The work done is quite inconclusive in terms of impact and clinical relevance.

The manuscript in my opinion is not suitable for the publication.

Specific points:

  1. In figure 1A there isn't the analysis in H460R cells. Why?

  2. Figure 1A, B: OVCAR3 presented the TCF4 gene deletion (Fig.1A) , but in RT-qPCR the expression of TCF4 mRNA in resistant cells didn't seem to have differences respect to sensitive cells. The authors should explain this discrepancy.

    The authors should show all the data they discuss, even the negative ones, in all four cell lines tha they have selected.

  3. Fig.2D: Also in this case, the authors showed the activation of WNT pathway in A2780 cells but they have to provide a a reason why the starting hypothesis was no longer valid in H63 cells despite having the TCF4 deletion.

  4. Up to figure 3, the authors try to validate with in vitro experiments their starting hypothesis only in a cell line of ovarian cancer (A2780). The title is misleading because it is indicated "Lung tumorigenesis".

  5. Fig.3: To address the effect of ITF2 on cisplatin resistance the authors should perform other assays as TUNEL, PARP and CASPASE3,7 cleavage, others.

  6. Figure 4 and Suppl Fig 2: Kaplan Meyer analysis was very confused. The authors have interrogated at least four cohorts of patients asking different things every time. Merely, KM analysis (OS and RFS) in TCGA for ITF2 and DKK1 expression. Why the authors analysed also HNSCC patients? Which is the rationale?

  7. About the reasons that I have expressed in point 4, the authors performed RNAseq in lung cancer patients without a strong background and preliminary data in vitro on cell lines. The experiments are conducted on ovarian carcinoma cells. It would have been more logical to make RNAseq in OvCa patients.

  8. Fig.6 the analysis on patients are confused.

  9. 25 NSCLC and 10 ovarian cancer samples are really too low.

Author Response

Answers to Reviewer 4

Point 1: In figure 1A there isn't the analysis in H460R cells. Why?

We found two common chromosomal regions deleted in the H23, A2780 and OVCAR3 cell lines after having established them resistant to cisplatin. These regions were 18q21.2-18q21.31 and 18q21.32. These deletions were not observed in the H460 cells. Since the initial objective of the manuscript was the evaluation of the effect of cisplatin, we focused on these three cell lines with common cytogenetic alterations after treatment with the drug.

In fact the authors agree in the interest of showing negative data, so we have included the next image of H460 in Supplementary Figure 1A, showing the absent of deletion in the described chromosomic area. We have included a reference to this figure in the main text in lines 82 and 89, and have changed the legend for Supplementary figure 1 in lines 574-581 as well as in the new Supplementary Figure1 as follows: “Supplementary Figure 1: (A) Picture extracted from the Agilent Cytogenomics 3.0.1.1 software showing the absence of ITF2 deletion in the H460R cells. (B) Picture extracted from the Agilent Cytogenomics 3.0.1.1 software showing the LRP1B deletion in chromosome 2 in H23R and H460R cell lines. (C) Relative mRNA expression levels of LRP1B measured by qRT-PCR. The results show the mean fold induction compared to the sensitive cells. Gene expression was normalized to GAPDH. Data represent the relative expression levels obtained from the combination of two independent experiments measured in triplicate ± SD. ns: not significant.

In addition, all the genomic information associated to the CGH array can be downloaded by the readers at the incorporated GEO repository number GSE129692, as indicated in line 466.

 The final version of the Supplementary Figure 1 would be as follows:

Point 2: Figure 1A, B: OVCAR3 presented the TCF4 gene deletion (Fig.1A) but in RT-qPCR the expression of TCF4 mRNA in resistant cells didn't seem to have differences respect to sensitive cells. The authors should explain this discrepancy. The authors should show all the data they discuss, even the negative ones, in all four cell lines tha they have selected.

In the OVCAR3 cell line, the deletion of ITF-2 was observed with 36% mosaicism, an alteration that was not enough to detect a significant decrease in gene expression as shown in Figure 1B. This fact was discussed in lines 343-348 of the manuscript as follows “ITF2 expression changes were confirmed in H23R and A2780R but not in OVCAR3R cells, also probably due to the level of mosaicism (36%) observed in these cells. Our results indicate that low levels of mosaicism would make the validations of expression changes by another quantitative technique difficult, probably because the alterations at expression levels are not significant enough to be detected”

In addition and following reviewer recommendations, we have first, provided the following data to the reviewer that supports our quantitative results in OVCAR3 cells. We are showing a comparison of TCF4 expression levels between sensitive and resistant cells. These data are extracted from the RNAseq data obtained in our cellular models. As we can see in the figure, the 30% of mosaicism observed in OVCAR3R cells does not correlate with a decrease in TCF4expression. Second, we have included the data associated with H460 cell lines, in which we could not find any deletion in both analyzed chromosomic regions: 18q21.2-18q21.31 and 18q21.32. as commented in the avowed concern. The negative data associated with LRP1B gene in the respective cell lines are also shown in Supplementary Figure 1.

Point 3: 2D: Also in this case, the authors showed the activation of WNT pathway in A2780 cells but they have to provide a a reason why the starting hypothesis was no longer valid in H63 cells despite having the TCF4 deletion.

Our data show that platinum treatment, frequently induces the indicated deletion (in three of the four cell lines treated) and that it affects the entire ITF2 gene, while in the cell line H460 this deletion was not observed. Knowing the function of this gene in the Wnt pathway, we wanted to investigate how this pathway could be affected by the deletion of ITF-2after treatment with platinum. However, it was impossible to assess changes in Wnt pathway in the H23 cell line because we observed a constitutive activity of bcat/TCF, circumstance that was in agreement with previous reports describing that the H23 cells have an increased expression of Wnt2, which causes Wnt/bcat activation in an autocrine manner. (Oncogene. 2009 May 28; 28(21): 2163–2172. doi:10.1038/onc.2009.82). This is why we consider that in H23 cells it will be hard to manipulate the pathway.

We have clarified this condition in the manuscript, lines 370-373 a follows: “In contrast, we did not observe differences in H23 cells, but a constitutive activity of bcat/TCF, circumstance that was in agreement with previous reports describing that the H23 cells have an increased expression of Wnt2, what causes Wnt/bcat activation in an autocrine manner [26 ]therefore, exogenous activation may not show a difference.”

Point 4: Up to figure 3, the authors try to validate with in vitro experiments their starting hypothesis only in a cell line of ovarian cancer (A2780). The title is misleading because it is indicated "Lung tumorigenesis".

One of the translational strengths of our manuscript is the high RNA quality from fresh-frozen tumor samples, mainly from patients harboring early stages of non-small cell lung cancer, which allowed us to perform a comprehensive transcriptome study to delve into the potential role of ITF2. For this reason we decided to emphasize the mention of lung cancer in the title. It was decided to include tumorigenesis in the title because the functional experiments of the Wnt pathway with TOP / FOP vectors, with and without ITF2overexpression, demonstrate modifications of the Wnt pathway and do not imply treatment with platinum. In addition in most of tumors analyzed both, our cohorts and in the "in silico" interrogated cohorts, showed lower levels of ITF2 expression than in normal tissues, before any chemotherapy treatment.

Nevertheless, we understand the reviewer's concern by including only one tumor type in the title when, throughout the manuscript, data related to other tumors of epithelial origin are presented. We therefore propose modifying the title of “A novel role for the tumor suppressor gene ITF2 in lung tumorigenesis and chemotherapy response”, by “A novel role for the tumor suppressor gene ITF2 in tumorigenesis and chemotherapy response”.

Point 5: 3: To address the effect of ITF2 on cisplatin resistance the authors should perform other assays as TUNEL, PARP and CASPASE3, 7 cleavage, others.

The authors agree with the fact of measuring cellular death by alternative methodologies would support and enrich our data correlating ITF2 overexpression and the increased cell mortality measured by Crystal Violet survival curves. However andunfortunately, due to the current situation derived from the COVID-19 in Madrid it is impossible for us to perform this experiment suggested by the reviewer. In our city Madrid, the alarm condition and the confinement situation at home has been activated, without being able to access the laboratory, right now for an indefinite time. We consider that this single experiment would not essentially induce substantial changes regarding the conclusions derived from the research, so we would like to appeal to the good judgment of the referee to allow us to respond to the rest of the concerns.

Point 6: Figure 4 and Suppl Fig 2: Kaplan Meyer analysis was very confused. The authors have interrogated at least four cohorts of patients asking different things every time. Merely, KM analysis (OS and RFS) in TCGA for ITF2 and DKK1 expression. Why the authors analysed also HNSCC patients? Which is the rationale?

We apologize if the presented data was confusing. Our initial analysis focused on lung and ovarian epithelial tumors, but since other epithelial tumors share common oncogenic drivers and chemotherapy-resistant mechanism, we explored if TCF4 deletion was present in other epithelial tumors and if could predict patient outcome. Head and neck SCC and esophageal carcinomas are both epithelial tumors that share expression signatures with lung squamous cancer, such as SOX2and TP63 amplifications, TP53 mutations and others. For that reason, we explored the predictive potential of TCF4 low expression, showing that ii can also predict overall survival in those patients. Further analysis will need to be done to test TCF4 function on those types of tumors, but the presented data suggest that TCF4 low expression could be used as a pan biomarker for epithelial tumors.

Point 7: About the reasons that I have expressed in point 4, the authors performed RNAseq in lung cancer patients without a strong background and preliminary data in vitro on cell lines. The experiments are conducted on ovarian carcinoma cells. It would have been more logical to make RNAseq in OvCa patients.

We really appreciate this valuable observation from the reviewer, so these authors will try to clarify why still consider there is enough rationale to perform RNAseq in our valuable early stages primary non-small cell lung cancer tumors.

As you may know, most of the already standardized human cancer cell lines have been established from metastatic origin. These cells are very aggressive and carry many genetic and cytogenetic alterations in comparison with the observed in primary tumors. That is the reason why it is hardly difficult to find human cancer cell lines sensitive to a drug in order to establish the resistant subclones. Then, it is rather complicated to establish cellular models resistant to cisplatin, as it can takes more than a year to have resistant subclones for each human cancer cell line by increasing the cisplatin doses in equilibrium with the cell mortality. In this manuscript, we have worked with eight paired sensitive/resistant human cancer cell lines established from lung and ovarian cancers established in our laboratory in the last 9 years. By using 2 cell lines for each tumor type the bioinformatics study associated with the CGH arrays had enough power to be performed. 

Since the initial objective of the manuscript was the evaluation of the cytogenetic alterations induced by cisplatin, we focused on the study of the three cell lines with common cytogenetic alterations after treatment with this drug, considering that the common genes would potentially be involved  to the acquired resistance to cisplatin. ITF2 gene was completely deleted in three resistant subclones and in two of them, the deletion correlated with a decreased expression, because in these two cell lines the mosaicism in the deletion was over 40%. We wound had like to test ITF2 role in both cell types lung and ovarian, but it was impossible to assess changes in Wnt pathway in the H23 cell line because, as we explained above, these cells harbor a Wnt/bcat activation in an autocrine manner.

Nevertheless our “in silico” analysis on 1,053 patients with NSCLC from the TCGA database identified the frequent loss of expression of the ITF2 gene (8.65% in adenocarcinomas and in 6.8% squamous cell carcinoma), validating what we observed when comparing tumor and non-pulmonary pathologies in the fresh samples collected in our laboratory. Furthermore, we analyzed lung human patient samples and could confirm the predictive value of our findings in vitro, that TCF4deletion and DKK1 over-expression can predict the outcome of these patients. Therefore, with all these compiled data, we could not rule out that the loss of ITF2could not affect the tumorigenic process in NSCLC patients just because in H23 cells it was not possible to perform the functional tests for Wnt activation; more if we consider that H23 cells are established from metastatic origin with a much more aggressive phenotype and carrying many genetic and cytogenetic alterations in comparison primary tumors.

 For this reason, we analyzed the transcriptome and the components of this pathway in primary tumors form NSCLC patients samples, for which we had an enough number of samples with high quality to do bioinformatic analysis, as previously described in response number 4,while carried out the functional tests of the Wnt pathway on the A2780 cell line, as it is a clear model in which the situation of resistance to cisplatin was concomitant with a deletion of ITF2 and an activation of Wnt.

We didn’t perform the RNAseq study in the ovarian cancer samples because we could only identify 10% of samples having an opposite expression between ITF-2 and DKK1, which prevented us to perform a bioinfomatic contrast to further explore the role of the Wnt-signaling pathway since they are based in an inverse expression of DKK1 and ITF-2. In addition,in our analysis of ovarian cancer from the TCGA cohort, we could only identify 13 patients with deletions of TCF4 or low expression, and 18 with an increased TCF4 expression out of 201, but there was no significant difference in overall survival between the two groups of patients.

We hypothesize that in ovarian cancer, the presence of low TCF4 level does not predict overall survival, but as we demonstrated, it could explain cisplatin resistance. Since the TCGA samples are from patient non treated, we could just speculate about its function in drug resistance based in our in vitro data, and further analysis will need to be done using samples from treated patients, which are incredibly hard to obtain.

Point 8: 6 the analysis on patients are confused.

We have tried to make the Figure 6 more comprehensive by improving the next explanation in the legend:

(D) Correlation between ITF2 and HOXD9 expression levels with the overall survival analyzed in NSCLC patients selected form the RNA-seq analysis. Left Y axis represents the quantitative expression levels of ITF2 (grey circles) measured by qRT-PCR as 2-∆∆Ct and its correlation with Overall Survival in days (grey line, r = 0.4). Right Y axis indicates the quantitative expression levels of HOXD9 (black triangles) measured by qRT-PCR as 2-∆Ct and its correlation with Overall Survival in days (black dotted-line, r = -0.4).

Point 9: 25 NSCLC and 10 ovarian cancer samples are really too low.

We agree with the comment of the reviewer. That’s why we explored the data sets available in the TCGA that include 566 Adenocarcinoma and 487 squamous cell carcinoma patient samples and validate the predictive value of TCF4/ITF-2, DKK1 and HOXD9.

Round 2

Reviewer 2 Report

The authors have addressed all my concerns. 

Reviewer 4 Report

The authors satisfactorily justified my requests and clarified some points. I would have required an extra effort and with the same data, that are now acceptable, I would have reorganized the manuscript. However in this version the manuscript is sufficiently explanatory without missleading.
I agree that in these historical circumstances two apoptotic assays, although more quantitative, only in this case where I still recognize that a lot of work has been done, do not alter the general results of the work.
The rebuttal letter was very convincing and well written, showing me that the authors still have mastery of the project despite the fact that the manuscript is sometimes out of focus.

In this version the manuscript is acceptable for the publication.